# Systems Biology Analysis Reveals Eight SLC22 Transporter Subgroups, Including OATs, OCTs, and OCTNs

**DOI:** 10.3390/ijms21051791

**Published:** 2020-03-05

**Authors:** Darcy C. Engelhart, Jeffry C. Granados, Da Shi, Milton H. Saier, Michael E. Baker, Ruben Abagyan, Sanjay K. Nigam

**Affiliations:** 1Department of Biology, University of California San Diego, San Diego, CA 92093, USA; dengelha@ucsd.edu; 2Department of Bioengineering, University of California San Diego, San Diego, CA 92093, USA; j6granad@ucsd.edu; 3School of Pharmacy and Pharmaceutical Sciences, University of California San Diego, San Diego, CA 92093, USA; das046@ucsd.edu (D.S.); rabagyan@ucsd.edu (R.A.); 4Department of Molecular Biology, Division of Biological Sciences, University of California San Diego, San Diego, CA 92093, USA; msaier@ucsd.edu; 5Department of Medicine, University of California San Diego, San Diego, CA 92093, USA; mbaker@ucsd.edu; 6Department of Pediatrics, University of California San Diego, San Diego, CA 92093, USA

**Keywords:** transporters, endogenous metabolism, functional subgroups, SLC22, remote sensing and signaling, drug transporters, gut microbiome, chronic kidney disease

## Abstract

The SLC22 family of OATs, OCTs, and OCTNs is emerging as a central hub of endogenous physiology. Despite often being referred to as “drug” transporters, they facilitate the movement of metabolites and key signaling molecules. An in-depth reanalysis supports a reassignment of these proteins into eight functional subgroups, with four new subgroups arising from the previously defined OAT subclade: OATS1 (SLC22A6, SLC22A8, and SLC22A20), OATS2 (SLC22A7), OATS3 (SLC22A11, SLC22A12, and Slc22a22), and OATS4 (SLC22A9, SLC22A10, SLC22A24, and SLC22A25). We propose merging the OCTN (SLC22A4, SLC22A5, and Slc22a21) and OCT-related (SLC22A15 and SLC22A16) subclades into the OCTN/OCTN-related subgroup. Using data from GWAS, in vivo models, and in vitro assays, we developed an SLC22 transporter-metabolite network and similar subgroup networks, which suggest how multiple SLC22 transporters with mono-, oligo-, and multi-specific substrate specificity interact to regulate metabolites. Subgroup associations include: OATS1 with signaling molecules, uremic toxins, and odorants, OATS2 with cyclic nucleotides, OATS3 with uric acid, OATS4 with conjugated sex hormones, particularly etiocholanolone glucuronide, OCT with neurotransmitters, and OCTN/OCTN-related with ergothioneine and carnitine derivatives. Our data suggest that the SLC22 family can work among itself, as well as with other ADME genes, to optimize levels of numerous metabolites and signaling molecules, involved in organ crosstalk and inter-organismal communication, as proposed by the remote sensing and signaling theory.

## 1. Introduction

The SLC (solute carrier) gene family includes 65 families with over 400 transporter genes. In humans, 52 of these families are expressed, encompassing more than 395 genes and it has been estimated that 2000 (10% of the genome) human genes are transporter-related [1]. Various solute carrier 22 (SLC22) members are expressed on both the apical and basolateral surfaces of epithelial cells where they direct small molecule transport between body fluids and vital organs, such as the kidney, liver, heart, and brain [2]. SLC22 transporters are also found in circulating cell types such as erythrocytes (e.g., SLC22A7), monocytes, and macrophages (e.g., SLC22A3, SLC22A4, SLC22A15, and SLC22A16) [3,4]. With recent calls for research on solute carriers, there has been a large influx of data over the past five years, including novel roles in remote sensing and signaling, leading to the need for a more comprehensive understanding of the functional importance of transporters [5].

The SLC22 family is comprised of at least 31 transporters and is found in species ranging from *Arabidopsis thaliana* of the plant kingdom to modern day humans [6,7]. Knowledge surrounding this family of proteins has expanded greatly since its proposed formation in 1997, when SLC22A6 (OAT1, originally known as novel kidney transporter or NKT) was first cloned [8]. Its homology to SLC22A1 (OCT1) and SLC22A7 (OAT2/NLT) led to the establishment of a new family (SLC22, TC# 2.A.1.19) of transport proteins within the major facilitator superfamily (TC# 2.A.1, MFS) as classified by the IUBMB-approved transporter classification (TC) system [8,9]. These proteins all share 12 α-helical transmembrane domains (TMD), a large extracellular domain (ECD) between TMD1 and TMD2, and a large intracellular domain (ICD) between TMD6 and TMD7 [10]. Research has shown these transporters to be integral participants in the movement of drugs, toxins, and endogenous metabolites and signaling molecules, such as prostaglandins, urate, α-ketoglutarate, carnitine, and cyclic nucleotides across the cell membrane [11].

As key players in small organic molecule transport, SLC22 members are hypothesized to play a role in the remote sensing and signaling theory [12,13,14,15]. The remote sensing and signaling theory posits that ADME genes—conventionally viewed as central to the absorption (A), distribution (D), metabolism (M), and elimination (E) of drugs, namely drug transporters and enzymes—aid in maintaining homeostasis through remote communication between organs via metabolites and signaling molecules in the blood that may in turn regulate gene expression [16]. This remote communication is supported by the example of serum uric acid levels. In the setting of the compromised kidney function, the increase in serum uric acid seems to be partly mitigated through a compensatory increase in the expression and/or function of ABCG2 in the intestine, which allows the excretion of uric acid in the feces rather than the urine [17,18]. Current research is focusing on determining the ways in which these transporters collaborate to regulate metabolite levels throughout the body [19].

Rather than maintaining a simple division of SLC22 into organic anion transporters (OATs), organic cation transporters (OCTs), and organic zwitterion/cation transporters (OCTNs), previous evolutionary studies have identified six phylogenetic “subclades”—OAT, OAT-like, OAT-related, OCT, OCTN-related, and OCTN—within the OAT and OCT “major clades” [10]. These subclades consist, on average, of three to four members with the exception of the OAT subclade that claims more than half of the 31 known members of SLC22 [10]. Although these subclades are phylogenetically sound, the endogenous functions of many SLC22 members within the six subclades remain ill-defined or unknown. With the emergence of new functional data, we performed a re-analysis of the SLC22 family to better characterize the functional, endogenous grouping of these transporters. Our re-analysis shows eight apparent subgroups, with four of these subgroups arising out of the previously defined (but very large) OAT subclade. Since these groupings are more closely related to well-known OATs rather than OCTs, OCTNs, or other subclades, we refer to these as OAT subgroups (OATS1, OATS2, OATS3, and OATS4).

We considered many factors in our re-analysis of SLC22 and subsequent designation of functionally based subgroups. To better describe the subgroups while still highlighting the nuances of each individual transporter, we utilized data from genomic loci, tissue expression, sequence similarity searches, proteomic motif searches, and functional transporter-metabolite data from GWAS, in vitro assays, and in vivo models. In place of phylogenetic studies, we performed multiple sequence alignments (MSA) and generated guide-trees that are based on sequence similarity or homology and thus provide more insight into function than solely phylogenetic studies. While the SLC22 family is composed of putative transporters, some members, like Slc22a20 and Slc22a17, have proposed mechanisms that differ from those of classic transporters [20,21]. To that effect, we explored the sequence similarities between SLC22 transporters and non-transport related proteins. We also used systems biology tools to develop an SLC22 transporter-metabolite networks as well as networks for each subgroup. This analysis elucidates the diversity of the endogenous functions of SLC22 transporters in various tissues and provides an updated functional framework for assigning each transporter to a subgroup. Considering the importance of SLC22 transporters, forming functional groups that incorporate endogenous substrates and tissue expression patterns can help better define their roles in intra-organ, inter-organ, and inter-organismal communication.

## 2. Results

Emerging data continue to indicate the centrality of the SLC22 family (particularly OATs, OCTs, and OCTNs) in endogenous physiology [5,16]. Our thorough reanalysis of the previously described phylogenetic subclades [10] revealed eight functional subgroups: OATS1, OATS2, OATS3, OATS4, OAT-like, OAT-related, OCT, and OCTN/OCTN-related (Table 1). By thus grouping this large family of proteins, we highlighted differences in substrate selectivity, showing that each member has a unique profile of associated metabolites. Based on the number of different metabolites it interacts with, each SLC22 transporter can be classified as relatively mono-, oligo-, or multi-specific. In what follows, publicly available data from GWAS, in vitro, and in vivo datasets were used to build functional networks that support the subgroups (Figure 1). In addition to these functional data and systems biology analyses, subgroups were also supported by structural, genomic, and other analyses explained below. Since some SLC22 members remain understudied, we also investigated low level sequence identity with non-transport proteins to better characterize these “orphaned” transporters.

In Figure 1 (in which metabolites linked to only a single transporter are not shown), 24 SLC22 proteins are linked to 79 unique metabolites, highlighting the physiological relevance of this family. This representation also brings attention to the number of shared substrates among SLC22, with 222 total edges present in this highly connected trimmed network. The multi-specific, oligo-specific, and mono-specific nature of different family members suggests how one transporter (e.g., a multi-specific member) may be able to compensate for the reduced function of another transporter (e.g., a mono-specific transporter). Furthermore, several of the metabolites interacting with the transporters (prostaglandins, carnitine derivatives, and bile acids) belong to different metabolic pathways, indicating that many processes, at both the systemic and cellular level, are dependent upon SLC22. In the following sections, the role of each subgroup in regulating metabolites in this larger SLC22 network was discussed in more detail.

### 2.1. Analysis of Substrate Specificity and Selectivity Helps Categorize Mono-, Oligo-, and Multi-Specificity of SLC22 Members

The concept of multi-, oligo-, and mono-specific SLC22 transporters was supported in part based on the number of unique drugs that are known to interact with each SLC22 member (Table 2) [22]. Several SLC22 members (e.g., OAT1 and OCT2) are best known as “drug” transporters and due to this association, many have been extensively tested as potential drug targets. While drugs were not the primary focus of this research, the number of drugs a transporter is linked to is indicative of how many structurally different substrates it can interact with. This may translate to endogenous compounds from different metabolic pathways. As interest in solute carriers has increased over the past decade, there has been a large influx of functional data (Appendix A). We used these data to validate our initial specificity assignments, and found that, for the most part, the metabolite data were in agreement with the drug data. A transporter linked to many unique drugs was often linked to many unique metabolites. For example, OATS1 members SLC22A6 and SLC22A8 are linked to 100 or more drugs, respectively. This is reflected in the metabolite data, as each transporter was associated with at least 50 unique metabolites, confirming their multi-specific nature. OATS4 members SLC22A9, SLC22A10, SLC22A24, and SLC22A25 are understudied with respect to drugs. As a group, they are only associated with three drugs, making it difficult to predict their substrate selectivity. Endogenously, the group appears to have relatively mono-specific members that are dedicated to conjugated sex steroids, and oligo-specific members, which are linked to conjugated sex hormones, short chain fatty acids, and bile acids.

### 2.2. Construction of Functional Networks from Metabolite-Transporter Interaction Data Support the Eight Subgroups

To visualize these transporter-metabolite interactions, which were acquired from a combination of GWAS, in vivo, and in vitro studies, we created networks using Cytoscape [23]. These networks allowed us to see the extent of unique and overlapping substrate specificity between transporters in the SLC22 family and within the proposed subgroups (Appendix A). The networks also provide, for the first time, a systems biology lens into the subgroup (as opposed to a single transporter) function. In these networks, all edges are undirected and represent a statistically significant result linking an SLC22 member to a metabolite. To give an example, the OATS1 network uses the members (SLC22A6, SLC22A8, and SLC22A20) as central nodes. Each associated metabolite is connected to the member, and the networks are then combined to represent the entire subgroup and demonstrate how a metabolite may be linked to multiple transporters (Appendix A). Functional data were available for 21 of 31 known SLC22 transporters. The trimmed SLC22 network is displayed in Figure 1, the individual subgroup networks are in Appendix A, and the total SLC22 network is in Appendix A. The compiled data with transporter, metabolite, study, quantitative metric, and citation are present in Appendix A.

While there is no single metabolite that is associated with all SLC22 transporters, some are linked to multiple family members, and thus may be a hallmark of the subgroup or family as a whole. These metabolites are prostaglandin E2, prostaglandin F2, estrone sulfate, uric acid, carnitine, and creatinine, which are each linked to at least five different SLC22 members, respectively (Appendix A). This result demonstrated that SLC22, as a group, is involved in regulating several metabolic processes, ranging from blood vessel dilation through prostaglandins to cellular energy production through carnitine [24,25]. This also implies that the particular structural features of the SLC22 family in general (12 TMD, large ECD between TMD1 and TMD2, and large ICD between TMD6 and TMD7) lend itself well to interacting with these compounds. This is further supported by the subgroup-specific network analyses and motif analysis we performed (Figure 1).

### 2.3. OATS1 (SLC22A6, SLC22A8, and SLC22A20) Handles a Wide Variety of Metabolites, Signaling Molecules, Uremic Toxins, and Odorants

Several metabolites have been identified as substrates of SLC22A6 (OAT1) and SLC22A8 (OAT3). While many are unique, there is notable overlap. Both OAT1 and OAT3 interact with uremic toxins (indoxyl sulfate, p-cresol sulfate, and uric acid) and gut microbiome derived products (indolelactate and 4-hydroxyphenylacetate), as well as many of the more general SLC22 metabolites, like prostaglandin E2, prostaglandin F2, uric acid, and creatinine [26,27,28,29,30]. SLC22A20 (OAT6), while not as well-studied, has affinity for several odorants and short chain fatty acids that are also associated with OAT1 [31]. OAT1 and OAT3 are clearly multi-specific, and OAT6 appears to be oligo-specific, as it handles both odorants and some short chain fatty acids. With respect to remote signaling, the shared metabolites among these transporters (Appendix A) were noteworthy because of their tissue localization (Table 3). OAT1 and OAT3 were primarily expressed in the kidney proximal tubule, with some expression in other tissues, like the choroid plexus and retina (Table 3). OAT6, however, is expressed in the olfactory mucosa of mice, presumably reflecting its affinity for odorants [21,31,32]. In the kidney, OAT1 and OAT3, along with many other SLC22 transport proteins, help regulate the urine levels of many metabolites and signaling molecules, which may potentially facilitate inter-organismal communication. For example, a volatile compound in one organism may be excreted into the urine through OAT1 and then somehow sensed by another individual of the same or different species through a mechanism involving OAT6 in the olfactory mucosa [12].

### 2.4. OATS2 (SLC22A7) is a Systemically-Expressed Transporter of Organic Anions and Cyclic Nucleotides

SLC22A7 (OAT2) is the only member of the OATS2 subgroup and is associated with prototypical SLC22 substrates, such as prostaglandins, carnitine, creatinine, and uric acid [29,35,36,37]. Evolutionarily, OAT2 appeared to be single member subgroup with a distinct branching pattern and single common ancestor within our generated guide trees (Figure 2, Appendix A). OAT2 was also linked to cyclic nucleotides and dicarboxylic acids, which when taken with the previous metabolites, created a unique profile worthy of its own subgroup (Appendix A) [38]. Another distinguishing feature of OAT2 was its tissue expression patterns (Table 3). While its expression in the liver and kidney are common to many SLC22 members, it has been localized to circulating red blood cells, where it may function in cyclic nucleotide transport [3]. Its expression in a mobile cell type and transport of cyclic nucleotides raises the possibility that it may act as an avenue for signaling.

### 2.5. OATS3 (SLC22A11, SLC22A12, and Slc22a22) Functions to Balance Uric Acid and Prostaglandins

In humans, SLC22A11 (OAT4) and SLC22A12 (URAT1) share only two substrates, uric acid and succinate (Appendix A) [39,40]. Uric acid is a beneficial metabolite in the serum as it is thought to be responsible for more than half of human antioxidant activity in the blood [41]. However, high levels of uric acid can be harmful and are associated with gout [42]. URAT1 is associated with very few metabolites and is best understood for its role in uric acid reabsorption in the kidney proximal tubule, making it relatively mono-specific [39]. OAT4, on the other hand, has been shown to transport prostaglandins and conjugated sex hormones in addition to uric acid, making it oligo-specific [43,44,45]. URAT1 is almost exclusively expressed in the kidney, and OAT4 is expressed in the kidney, placenta, and epididymis (Table 3). The more diverse tissue expression of SLC22A11 seems consistent with its wider range of substrates. The subgroup differs in rodents because mice do not express Oat4. Instead, the rodent subgroup is composed of Slc22a12, known as the renal-specific transporter (Rst) in mice, and Slc22a22, known as the prostaglandin-specific organic anion transporter (Oat-pg). While Rst and Oat-pg do not share substrate specificity, together, they combine to play the role of URAT1 and OAT4 by handling uric acid and prostaglandins [46].

### 2.6. OATS4 (SLC22A9, SLC22A10, SLC2A24, and SLC22A25) Members are Specifically Associated with Conjugated Sex Hormones

GWAS analyses support the association of all human members of this subgroup with one common metabolite, etiocholanolone glucuronide, a conjugated sex hormone, with a *p*-value of 4.12 × 10^−27^ or lower for all members (Table 3, Appendix A) [47]. While this group shares at least one conjugated sex hormone, SLC22A24 and SLC22A9 appear to be more oligo-specific transporters, with SLC22A9 linked to short chain fatty acids and SLC22A24 linked to bile acids [48,49]. SLC22A10 and SLC22A25 are only linked to conjugated sex hormones, making them relatively mono-specific transporters (Appendix A) [47]. In terms of tissue expression, there is a distinct correlation between patterns and shared function amongst human OATS4 members (Table 4). We predicted that all four members are conjugated sex steroid transporters with SLC22A9, A10, and A25 showing high expression in the liver where conjugation of glucuronides and sulfates to androgens and other gonadal steroids occurs [48]. SLC22A24 has low expression levels in the liver but is highly expressed in the proximal tubule, where it is predicted to reabsorb these conjugated steroids [48]. This subgroup also includes a large rodent-specific expansion, consisting of Slc22a19 and Slc22a26-30. Although the rodent-specific expansion is greatly understudied, transport data for rat Slc22a9/a24 show shared substrate specificity for estrone sulfate with SLC22A24, but not for bile acids or glucuronidated steroids, which is consistent with the lack of glucuronides in rat urine and serum [48]. While sulfatases are extremely highly conserved amongst humans, rats, and mice, the separation of rodent- and nonrodent-specific OATS4 groups may be due to the species differences in expression and function of glucuronidases [50]. Despite their distinct differences from human OATS4 members in sequence similarity studies and minimal functional data, the rodent-specific transporters are also highly expressed in both liver and kidney [51].

### 2.7. OAT-Like (SLC22A13 and SLC22A14) has Potentially Physiologically Important Roles

Very little functional data are available for the OAT-like subgroup. SLC22A13 (OAT10/ORCTL3) has been well characterized as a transporter of both urate and nicotinate, but SLC22A14 has no available transport data [52]. However, Nʹ-methyl nicotinate is increased in the plasma levels of self-reported smokers, and GWAS studies have implicated SNPs in the SLC22A14 gene to be associated with success in smoking cessation [53,54]. Although these data do not directly relate SLC22A14 to nicotinate, it suggests a possible route of investigation into the functional role of this transporter, one that may, in some ways, overlap with that of OAT10. SLC22A13 is primarily expressed in the kidney, and although we found no human protein expression data for SLC22A14, transcripts for this gene are found at low levels in the kidney and notably high levels the testis (Table 4), which is in concordance with its critical role in sperm motility and fertility in male mice [55]. Future studies are required to determine the functional classification of this subgroup; however, our genomic localization and sequence-based analyses provided enough data to support the notion that these two belonged in their own individual subgroup.

### 2.8. OAT-Related (SLC22A17, SLC22A18, SLC22A23, and SLC22A31) is Anomalous Amongst SLC22 Members but has Interesting Functional Mechanisms and Disease Associations

The OAT-related subgroup was an outlier within the SLC22 family, consisting of the orphan transporters SLC22A17, SLC22A18, SLC22A23, and SLC22A31. SLC22A17 and SLC22A23 were strongly related, with greater than a 30% shared amino acid identity. When these two transporters were initially identified together as BOCT1 (SLC22A17) and BOCT2 (SLC22A23), it was noted that they both show high expression levels in the brain, as well as a nonconserved amino terminus that may negate prototypical SLC22 function [56]. SLC22A17 is known as LCN2-R (Lipocalin receptor 2) and is reported to mediate iron homeostasis through binding and endocytosis of iron-bound lipocalin, as well as exhaustive protein clearance from the urine as shown by high affinities for proteins such as calbindin [20,57]. SLC22A23 has no confirmed substrates, but SNPs and mutations within this gene have medically relevant phenotypic associations such as QT elongation, inflammatory bowel disease, endometriosis-related infertility, and the clearance of antipsychotic drugs [58,59,60]. SLC22A31 is the most understudied transporter of the SLC22 family but has been associated with right-side colon cancer [33]. SLC22A18 remains an outlier and lacks the characteristic SLC22 large ECD. Its membership within the SLC22 family is arguable due to high sequence similarity with the DHA H^+^-antiporter family (Appendix A) [10]. Further study is required to confirm if the OAT-related members share substrates as a group or if their sequence diversity and deviations from classical physical SLC22 member characteristics are the reason for their phylogenetic association.

### 2.9. OCT (SLC22A1, SLC22A2, and SLC22A3) Members Are Characteristic Organic Cation Transporters with High Affinities for Monoamine Neurotransmitters and Other Biologically Important Metabolites and Signaling Molecules

The OCT subclade of SLC22A1 (OCT1), SLC22A2 (OCT2), and SLC22A3 (OCT3) has ample data to support its formation and has been widely accepted and utilized as the prototypical subgroup of organic cation transporters. All three members of this subgroup transport monoamine neurotransmitters, carnitine derivatives, creatinine and the characteristic OCT substrates, MPP+, and TEA (Appendix A) [34,37,61,62,63,64]. All three members of this subgroup were expressed in the liver, kidney, and brain (Table 4). When considered together with the transport of neurotransmitters, this subgroup serves as an example of inter-organ communication between the brain and the kidney–liver axis via transporters. The systemic levels of these neurotransmitters and thus, their availability to the brain can be regulated by the expression of OCT subgroup members in the liver, where the metabolites can be enzymatically modified, and expression in the kidney, which may serve as an excretory route [7].

### 2.10. OCTN/OCTN-Related (SLC22A4, SLC22A5, SLC22A15, and SLC22A16) Subgroup Consists of Prototypical Carnitine and Ergothioneine Transporters

The OCTN/OCTN-Related subgroup is a combination of two previously established subclades, OCTN and OCTN-related [10]. Previous studies have mistakenly named SLC22A15 as CT1 (carnitine transporter 1), but this name actually belongs to SLC22A5 (OCTN2) [13]. GWAS data show that SLC22A4 (OCTN1), SLC22A5 (OCTN2/CT1), and SLC22A16 (FLIPT2/CT2) are heavily linked to carnitine and its derivatives [37]. This is consistent with in vitro data showing that OCTN2 and FLIPT2 are carnitine transporters [65,66]. Although OCTN1 has lower affinity for carnitine than OCTN2 and FLIPT2, it has high affinity for the endogenous antioxidant ergothioneine, which GWAS data suggest may be a shared metabolite with both SLC22A15 (FLIPT1) and FLIPT2 (SLC22A16; Figure 2B) [37,67]. SLC22A15 is associated with many complex lipids that are not characteristic of any other SLC22 transporter [47]. Although data are very limited, this anomalous SLC22 member so far appears to only share one potential substrate with this subgroup, but its inclusion is supported by multiple sequence alignments focusing on the intracellular loop and tissue expression patterns. Most other subgroups in this family are limited to a few tissues, mainly the liver and kidney, but the members of the OCTN/OCTN-Related subgroup are all expressed in at least five tissues as well as circulating immune cells (Table 4) [4,7]. This broad tissue expression pattern, in conjunction with our network analysis, supports the notion that these transporters’ main task is transporting carnitine derivatives, as carnitine metabolism is an energy producing mechanism in nearly every cell. It may also play a role in regulating levels of the antioxidant ergothioneine, which appears to be a unique substrate of this subgroup [24,68].

### 2.11. Multiple Sequence Alignment Further Supports the Classification of Subgroups

Our new subgroupings are primarily based on the endogenous function of the transporters, but they are also supported by additional analyses. These analyses are necessary, as structural and evolutionary similarities can predict functional traits that have yet to be discovered. Though the previously established phylogenetic subclades remain sound, our re-analysis includes new and updated amino acid sequences that support the proposed subgroups with more confidence, especially when investigating similarities within functional regions [10]. MSA programs were favored over phylogenetics because MSA searches are based upon structural similarities rather than evolutionary relatedness [69]. These structural similarities, especially in the large ECD (extracellular domain) and large ICD (intracellular domain) of SLC22 proteins, may indicate shared function.

Full length sequence analysis via Clustal-Omega, MAFFT, and ICM-Pro v3.8-7 supported the division of SLC22 into eight subgroups (Figure 2A, Appendix A). While the OATS1, OATS2, OATS4, OAT-like, OAT-related, and OCT subgroups were supported by full-length sequence analyses, OATS3 and OCTN/OCTN-Related required a more rigorous investigation. To further clarify “borderline” subgroup assignments from the full-length sequence analysis, sequence similarity between the ECDs and ICDs of all human and mouse SLC22 members was determined using ICM-Pro v3.8-7, and the results were visualized via guide trees (Figure 2B,C). ECD alignment preserved all eight subgroups, with the exception of SLC22A15 in the OCTN/OCTN-Related subgroup. In contrast, ICD alignment preserved only the OATS4, OATS2, and OCT subgroups.

The branching pattern of OATS3 member Oat-pg (Slc22a22) differs between tree variations. These analyses consistently indicate a similar relationship between Oat-pg and OATS3, as well as OATS4. However, in an analysis of the SLC22 ECDs, it is most closely associated with OATS3 over any other subgroup. This, in conjunction with shared substrate specificity with both SLC22A12 and SLC22A11, and not OATS4 members, supports its membership within the OATS3 subgroup [29,39,40,46].

In full-length sequence alignments, the grouping of SLC22A4, SLC22A5, and Slc22a21 is consistently conserved, while the topology of both SLC22A15 and SLC22A16 is irregular. Despite this, analysis of the large ECD shows similarity between all OCTN/OCTN-related members other than SLC22A15. Previous analyses have noted the large difference between the ECD of SLC22A15 and all other SLC22 members, which is supported by our analysis in Figure 2B [10]. Interestingly, there appears to be some similarity between the large intracellular domains of SLC22A16 and SLC22A15. Although much of the support for the establishment of the OCTN/OCTN-related subgroup comes from functional data (Appendix A), the described MSA analyses highlight shared structural, and possibly functional, regions.

### 2.12. Analysis of Genomic Localization Highlights Evolutionary Relatedness of Subgroup Members and Suggests Basis of Coregulation

Genomic clustering within the SLC22 family has been previously described [10]. Specifically, genes found in tandem on the chromosome, such as OAT-like members SLC22A13 and SLC22A14, are predicted to have arisen from duplication events, indicating a strong evolutionary relationship. Despite the majority of the OAT subclade being found on chromosome 11 in humans and chromosome 19 in mice, clustering within the chromosome supports the division of the OAT subclade into smaller subgroups. For example, OATS4 members SLC22A9, SLC22A10, SLC22A24, and SLC22A25 appear in tandem on human chromosome 11 within the UST (Unknown Substrate Transporter) region of the genome. This region is analogous to the UST region within the mouse genome on chromosome 19, where the mouse-specific OATS4 members Slc22a19, Slc22a26, Slc22a27, Slc22a28, Slc22a29, and Slc22a30 reside as well as the rat UST region on chromosome 1 that contains Slc22a9/a24, Slc22a9/a25, Ust4r, and Ust5r (Table 3) [70,71]. It has been proposed that genes within clusters, to some degree, are coordinately regulated and thus are predicted to have similar overall tissue expression patterns [70,72,73]. Support for shared regulatory mechanisms of subgroup members within genomic clusters can be inferred from similar patterns of tissue expression or by expression of subgroup members along a common axis of metabolite transport such as the gut–kidney–liver axis. Genomic localization from the UCSC Genome browser and resultant tissue expression patterns for all SLC22 members are shown in Table 3.

### 2.13. Analysis of OAT Subgroup Specific Motifs Highlight Patterns Potentially Involved in Specificity

Motif analyses revealed subgroup specific motifs within functionally important regions, such as the large ICD, large ECD, and the region spanning TMD9 and TMD10, for all novel OAT subgroups [10,74]. However, the number of unique residues appears to be correlated to the range of substrate specificity.

Of the newly proposed OAT subgroups, OATS2 claims the smallest number of subgroup-specific amino acid motifs and is the only subgroup without a specific motif in TMD9 (Figure 3B). The lack of multiple subgroup-specific regions is interesting not only because this subgroup consists of a single transporter but also because this may be indicative of a more promiscuous transporter with a wide range of substrates, which is substantiated by the functional data, as described earlier in “*OATS2 (SLC22A7) is a Systemically-Expressed Transporter of Organic Anions and Cyclic Nucleotides*”. This pattern is also seen in OATS1, which consists of multi- and oligo-specific transporters OAT1, OAT3, and OAT6. In addition to having few subgroup-specific motifs, the multi/oligo-specific nature of this subgroup is reflected by the shared evolutionary conservation of the large extracellular domain with other OAT subclade members (Figure 3A).

To further clarify the membership of Oat-pg in OATS3, evolutionarily conserved motifs were determined between all three members, as well as just Slc22a11 and Slc22a12. This analysis revealed a total of ten evolutionarily conserved amino acid motifs between all three members, eight of which were present in the analysis of only OAT4 and URAT1 (Figure 3, Appendix A). Specifically, both analyses exhibited a notably large motif in the large intracellular loop found at D313-Q332 on URAT1 and Q312-G331 on OAT-PG (Figure 3C,D). This larger number of conserved regions seems consistent with a more limited range of substrates (e.g., uric acid and prostaglandins) [43].

Motif analysis was performed separately on the OATS4 rodent and non-rodent specific subgroups and the entirety of the OATS4 subgroup members. In all analyses, OATS4 claims the largest number of evolutionarily conserved and subgroup-specific amino acid residues amongst the OAT subgroups, supporting selective substrate specificity, possibly for conjugated sex steroids (Figure 3E,F). In the case of non-rodent transporters, a unique motif spanned the sixth extracellular domain and TMD12. This region is predicted to govern substrate specificity of transporters of the MFS, to which the SLC22 family belongs [74]. Recent publications defining the substrate specificity of SLC22A24 point to a more narrow range of substrates and conservation of this specific region amongst OATS4 members may explain the association of conjugated steroid hormones with SLC22A9, SLC22A10, SLC22A24, and SLC22A25 in GWAS studies [47,48]. Although further analysis was required to fully understand the relationship between the structure and substrate specificity in SLC22 transporters, we provided a basis for investigation into specific regions that might determine functional patterns. The sequences and *p*-values for each motif are in Appendix A.

### 2.14. Sequence Similarity Study Suggests Novel Potential Functions to Explore and Possible Tertiary Structure of SLC22

Each SLC22 member is a putative transporter, but there is evidence that suggests some members may have alternative mechanisms of action [31,57]. To further explore this possibility and to potentially find sequence similarity to other proteins, the specific amino acid sequences for the extracellular and intracellular loops of each SLC22 member were compared to all proteins in the ICM-Pro v3.8-7c database. The large extracellular loop of the OCT subclade (hSLC22A1-A3) showed notable homologies to human, cow, mouse, and rat SCO-spondin, a glycoprotein secreted by the subcommissural organ in the brain. In all of these species, SCO-spondin contains two potent binding sites for glycosaminoglycan (BBXB) and cytokines (TXWSXWS) as well as LDL receptor type A repeats. Human SCO-spondin shares 28.97% (pP = 5.47) and 27.43% (pP = 5.33) sequence identity with human SLC22A1 ECD and SLC22A3 ECD, respectively. The extracellular loop of mouse Slc22a16 shares 26% sequence identity (pP = 5.4) with chicken beta-crystallin B3 (CRBB3). Beta-crystallin is a structural protein mainly comprised of beta sheets [75]. The similarity between the ECD of mouse Slc22a16 and CRBB3 could point to potential for a beta sheet like configuration. Since none of the SLC22 family members have been crystallized, any insight into tertiary structure is of interest.

SLC22A31, a member of the divergent OAT-Related subclade, is the most ambiguous member of the SLC22 family with no functional data available. An investigation of the human SLC22A31 large ECD shows at least 30% shared sequence identity with RNA-binding protein 42 (RBM42) in mouse, rat, cow, and human. This analysis also showed a 37% sequence identity (pP = 5.5) shared between the ECD of hSLC22A31 and human heterochromatinization factor BAHD1. These and other interesting sequence similarities to proteins, including those involved in signaling, are noted in Table 5.

## 3. Discussion

In the years following the establishment of the previous SLC22 subclades, there has been a notable increase in functional data, particularly with respect to endogenous substrates, concerning these transporters and their substrates [10]. With these data, we are now in a position to better characterize the biology of these transporters, which play important physiological roles and are implicated in certain diseases. However, our newly proposed subgroups are not entirely dependent on functional data, as we have considered multiple approaches including phylogenetics, multiple sequence alignments, evolutionarily conserved motifs, sequence homology, and both tissue and genomic localization. Each of these approaches has individual value in that they reveal unique characteristics of each transporter; yet it is the combination of multiple approaches that ensures the full variety of available data (though still incomplete) for these transporters is considered when forming functional subgroups. We support the subgroups with a thorough literature search of metabolites associated with SLC22 proteins.

Although the functional data were inherently biased due to the high level of interest in some SLC22 members, particularly the “drug” transporters OAT1, OAT3, OCT1, and OCT2, for the majority of the transporters, there are enough data to create functional subgroups that play distinct and overlapping roles in metabolism (Figure 1, Appendix A). Genomic localization reveals evolutionary information and provides insight on how genes may arise from duplication events. Phylogenetic analysis determines the evolutionary relatedness of these proteins, while MSA, motif analysis, and sequence homology focus on structural similarities, which can be indicative of function. We often see that members of a subgroup are expressed in the same tissues or along functional axes. For example, substrates transported from the liver via SLC22 transporters (e.g., SLC22A1/OCT1) can be either excreted into or retrieved from the urine by other SLC22 members (SLC22A2/OCT2) of the same subgroup. Establishment of these functional subgroups may also inform future virtual screenings for metabolites of understudied transporters.

Protein families are established based on shared ancestry and structural similarity, which is commonly considered grounds for shared functionality. This is exemplified amongst SLC22 members with the generally shared structural characteristics of 12 TMDs, a large extracellular loop between TMD1 and TMD2, and a smaller intracellular loop between TMD6 and TMD7. Despite these shared features, we show here that there are many functional differences between these transporters. Although our analyses mostly align with previous evolutionary studies when considering ancestry, here, we show that phylogenetic grouping is not always reflective of a similar structure and function. For example, although the previously established OCTN subclade of SLC22A4, SLC22A5, and Slc22a21 does not share common ancestry with Slc22a16, the newly proposed group shares functional similarity and ECD homology. Thus, by expanding our investigation beyond phylogenetic relationships, we can now more appropriately group proteins from the same family and better understand their roles in endogenous physiology.

An important concept in the remote sensing and signaling network is that of multi-specific, oligo-specific, and relatively mono-specific transporters working in a coordinated function [16]. Multi-specific transporters are able to interact with a wide variety of structurally different compounds, oligo-specific with a smaller variety, and relatively mono-specific transporters are thought to interact with only one or a few substrates. Existing functional data suggest that it is unlikely that any truly mono-specific transporters exist within the SLC22 family, yet the different subgroups we have formed imply that multi-specific, oligo-specific, and relatively mono-specific transporters are more likely to form subgroups with transporters that share substrate specificity. Multi-specific transporters, like those in the OATS1 and OCT subgroups, handle a diverse set of drugs, toxins, endogenous metabolites, and signaling molecules [14,61]. Conversely, the OATS4 subgroup appears to be a collection of relatively mono-specific transporters with an affinity for conjugated sex steroid hormones, specifically etiocholanolone glucuronide, which is also supported by a recent study focused on SLC22A24, a member of the OATS4 subgroup [47,48]. Previous evolutionary studies have suggested that multi-specific transporters arose before the mono-specific transporters [10]. As evolution has progressed, more specific transporters have developed to handle the burden of changing metabolism. The multi-specific transporters have been more extensively characterized because of their importance in pharmaceuticals, but in the case of endogenous metabolic diseases, the oligo and mono-specific transporters may be more appropriate targets for drugs or therapies.

One of the best examples of multi-specific transporters working in concert with oligo, and mono-specific transporters is the regulation of uric acid [17,18]. Handling of uric acid mainly occurs in the kidney, but when renal function is compromised, multi-specific transporters regulate their expression to compensate. Two proteins, SLC22A12 (URAT1) and SLC2A9, are expressed in the proximal tubule and are nearly exclusively associated with uric acid. The multi-specific transporters SLC22A6 (OAT1) and SLC22A8 (OAT3) are also present in the proximal tubule and are able to transport uric acid. When the kidney is damaged, one would expect serum uric acid levels to increase because most of the proteins involved in its elimination are in the kidney. However, this is partly mitigated due to the increased expression of ABCG2 and/or functional activity in the intestine [17,18]. SLC2A9 is a relatively mono-specific transporter and ABCG2 (BCRP) is a multi-specific ABC transporter, and other uric acid transporters can be considered oligo-specific (e.g., SLC22A11). The example of uric acid serves to illustrate how, when certain mono-, oligo-, and multi-specific transporters are unable to perform their primary function, multi-specific transporters of the same or different function (even of the ABC superfamily) can use their shared substrate specificity to mitigate the consequences. It is generally assumed that all SLC22 family members are transporters. However, Slc22a17, a member of the outlier OAT-related subclade, functions as an endocytosed iron-bound lipocalin receptor and some SLC22 members have been suggested to function as “transceptors” due to homology with GPCR odorant receptors and shared odorant substrates [20,21]. Thus, to better understand the SLC22 family members’ individually unique functions and their placement into subgroups/subclades, we compared the full-length amino acid sequences, large ECDs, and large ICDs of all SLC22 family members to a database of known proteins.

When considering such a large number of proteins, the function on both local and systemic levels of metabolites is likely to be impacted. The SLC22 family is a central hub of coexpression for ADME (absorption, distribution, metabolism, excretion)-related genes in non-drug treated conditions, which underscores their importance in regulating endogenous metabolism through the transport of small molecules [16]. In the context of the remote sensing and signaling theory (RSST), it is essential to understand substrate specificity of different SLC22 members and the eight subgroups.

The RSST proposes that a network of ADME genes (drug transporters, drug metabolizing enzymes, and various regulation proteins) regulates the levels of hundreds if not thousands of small organic molecules with “high informational content” including key metabolites and signaling molecules involved in intra-organ, inter-organ, and inter-organismal remote communication. The RSST would seem to imply that organisms are constantly solving a multi-objective optimization problem, where balancing each particular compound’s serum concentration represents a single objective. Each compound present in the blood has a range of healthy concentrations, and when the concentration is outside of that range, the body must address it, in part through the regulation of transporters and enzymes. Due to their wide range of tissue expression and diverse functional roles at body fluid interfaces, the particular combination of transporters and enzymes are critical variables necessary for solving this multi-objective optimization problem. Transporters regulate the entry and exit of substrates to and from cells, but enzymes are responsible for the altering of these compounds. To use a simple hypothetical example, if a metabolite’s serum concentration is too high, a transporter with high affinity for that metabolite can move it into the cell, where an enzyme with high affinity for the substrate can change it so that it may re-enter the circulation or be more readily cleared from the body. The existence of abundant multi-specific and oligo-specific transporters and enzymes, in addition to relatively mono-specific ones, expressed differentially in tissues and at body fluid interfaces, allows for a highly flexible and responsive complex adaptive system that not only maintains homeostasis in blood, tissue, and body fluid compartments (e.g., cerebrospinal fluid), but also helps restore it after acute or chronic perturbations.

Thus, together, transporters and enzymes have tremendous potential to manage levels of metabolites, signaling molecules, and antioxidants in the circulation and in specific tissues. By developing functional groupings for the SLC22 family, we could better understand the metabolic networks in which they function and how their expression is utilized to regulate concentrations of metabolites, signaling molecules (e.g., cyclic nucleotides, prostaglandins, short chain fatty acids, and sex steroids), antioxidants (ergothioneine, uric acid), and other molecules affecting diverse aspects of homeostasis (e.g., lipocalin). Although this analysis focuses on the SLC22 family, a similar approach can be applied to develop a deeper understanding of other families of transporters and enzymes.

In the past, the majority of functional data have come from transport assays using cells overexpressing a specific SLC22 transporter and a single metabolite of interest. These assays lack uniformity and, as the OAT knockouts have shown, are not necessarily reflective of endogenous physiology [26,28,30]. Recently, GWAS studies have linked many metabolites to polymorphisms in SLC22 genes, and in vivo metabolomic studies using knockout models have also identified several metabolites that may be substrates of transporters [26,28,30,47]. In upcoming years, the integration of multiple types of omics data related to SLC22 family members with functional studies of transporters and evolutionary analyses will likely produce a more fine-grained picture of the roles of these and other transporters in inter-organ and inter-organismal remote sensing and signaling.

## 4. Materials and Methods

### 4.1. Data Collection

SLC22 human and mouse sequences were collected from the National Center for Biotechnology Information (NCBI) protein database. Sequences were confirmed and genomic loci were recorded using the University of California Santa Cruz (UCSC) genome browser by searching within each available species on the online platform (https://genome.ucsc.edu/cgi-bin/hgGateway) [76]. The NCBI BLASTp web-based program was used to find similar sequences to those from the NCBI protein database. BLASTp was run with default parameters using query SLC22 sequences from human or mouse. The database chosen was “non-redundant protein sequence” (nr) and no organisms were excluded [77]. Tissue expression of all human SLC22 members was collected from the Human Protein Atlas, GTEx dataset, Illumina Body Map, ENCODE dataset, and RNA-seq datasets available on the EMBL-EBI Expression Atlas (https://www.ebi.ac.uk/gxa/home) [78]. Tissue expression data were also collected via extensive literature search.

### 4.2. Sequence Alignment and Guide-Trees

Sequences for SLC22 were aligned using Clustal-Omega with default parameters via the online platform provided by the European Bioinformatics Institute (EMBL-EBI) (https://www.ebi.ac.uk/Tools/msa/clustalo/), as well as MAFFT (multiple alignment using fast Fourier transform) and ICM-Pro v3.8-7c [79,80,81,82]. Clustal-Omega, MAFFT, and ICM-Pro v3.8-7c produced similar topologies. These alignments were then visualized using The Interactive Tree of Life (http://itol.embl.de/) [83]. Topology was analyzed by branch length values, which are a result of the neighbor-joining method. This method calculates the number of amino acid changes between each organism and the common ancestor from which it branched. It then adopts the minimum-evolution criteria (ME) by building a tree, which minimizes the sum of all branch lengths to visually display relatedness [84]. SSearch36 was utilized to compare representative sequences of all members of the Drug:H+ Antiporter-1 (12 Spanner) (DHA1) Family (2.A.1.2) and the SLC22A family (2.A.1.19) from the Transporter Classification Database (http://www.tcdb.org/) with the Cyanate Porter (CP) Family (2.A.1.17) as an outgroup to further investigate the belongingness of SLC22A18 in either the SLC22 or DHA1 family [9,85]. SSearch36 is an exhaustive comparison method that uses the Smith–Waterman (SW) algorithm to compare FASTA files find sequence similarities [85,86].

ICM-Pro v3.8-7c was used to align sequences in FASTA format as well as perform homology searches of all human and mouse SLC22 sequences against ICM-Pro’s curated database of all known proteins [82]. Threshold for homology significance was determined by the probability of structural insignificance (pP), defined as the negative log of the probability value of a homology comparison. Alignments were discarded if the pP value was less than 5.0, indicating that the homology shared between two sequences is likely not due to random sequence similarities.

### 4.3. Motif Analysis

Motif comparisons were performed on the subgroups of the OAT subclade using the multiple expectation-maximum for motif elicitation (MEME; http://meme-suite.org/tools/meme) suite [87]. A threshold of 20 motifs containing a range of 6–20 amino acid length was set with the normal discovery mode. This detection method yielded a set of evolutionarily conserved motifs within all OAT subclade sequences (*n* = 57) as well as a set of evolutionarily conserved motifs for each of the four proposed OAT subgroups. These motifs were then mapped onto 2D topologies of one member from each of the newly proposed OAT subgroups (SLC22A6 for OATS1, SLC22A7 for OATS2, SLC22A12 for OATS3, and SLC22A9 for OATS4). A separate motif analysis was also performed for the rodent expansion consisting of Slc22a19, and Slc22a26-30 and was mapped onto mouse Slc22a27. Transmembrane domains (TMDs) of these transporters were predicted by the constrained consensus topology prediction server (CCTOP; http://cctop.enzim.ttk.mta.hu/) [88]. TMD locations and the motif locations were entered into TOPO2 (http://www.sacs.ucsf.edu/TOPO2/) to visualize the 2D representation of the transporters with the OAT subclade’s evolutionarily conserved motifs shown in blue and each subgroup’s evolutionarily conserved motifs shown in red [89].

### 4.4. SNP, Mutation, In-Vitro, Knockout, and Drug Transport Data

To determine the diversity of substrate specificity, the number of drugs that list SLC22 members as a target on DrugBank were recorded [22]. The Metabolomics GWAS server was utilized to determine SNPs within all SLC22 members. The dataset produced by Shin et al. (2014) with the cohort KORA+TwinsUK (blood) and the association of single metabolites was chosen. This dataset was searched by gene symbol (e.g., SLC22A6) [37,90]. The EMBL GWAS Catalog and Metabolomix’s table of all published GWAS with metabolomics (http://www.metabolomix.com/list-of-all-published-gwas-with-metabolomics/) were also utilized in searching for SNPs and their effect on metabolite transport by SLC22 members [91]. Current literature available on the NCBI gene database under gene references into functions (Gene RIFs; https://www.ncbi.nlm.nih.gov/gene/about-generif) was used to search for non-synonymous mutations that did not affect protein expression yet affected transport of metabolites and/or drugs. These methods were accompanied by an extensive literature search for in-vitro transport and knockout data. Most in-vitro data come from tissue culture assays from a variety of cell lines while most in-vivo data comes from genetic or chemical knockout mice. Metabolite data were abstracted from the aforementioned databases and confirmed via literature review. The import from table feature on Cytoscape 3.7.2 was used to generate functional networks for the entire SLC22 family and the subgroups [23]. A spring embedded layout was applied to the networks and the subgroups were color coded manually.

## Figures and Tables

**Figure 1 ijms-21-01791-f001:**
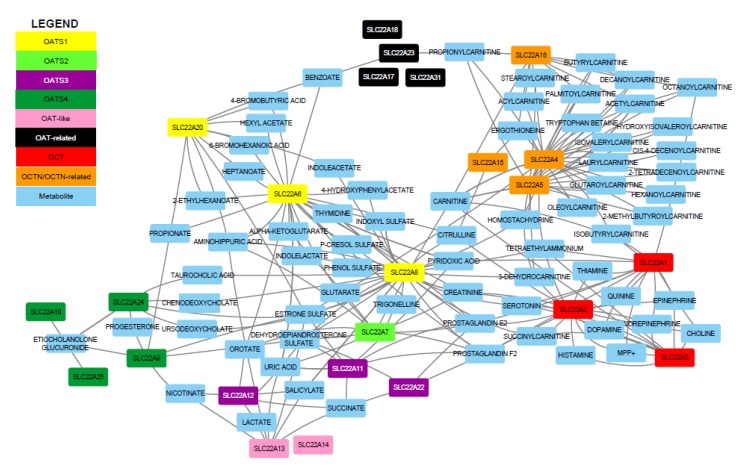
Pruned SLC22 network. All SLC22 transporters with functional data were initially included. Metabolites associated with only one transporter were removed for improved visualization. SLC22 transporters and metabolites are colored nodes. Each edge represents a significant transporter-metabolite association. Multiple edges connecting one metabolite to a specific transporter were bundled (e.g., in vitro and GWAS support).

**Figure 2 ijms-21-01791-f002:**
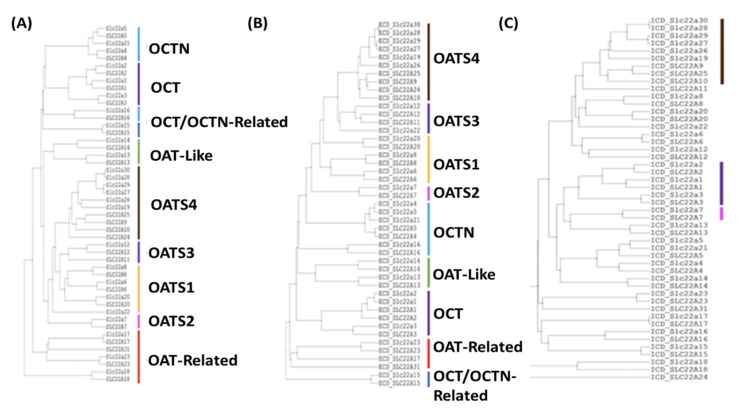
Multiple sequence alignment using ICM-Pro v3.8-7c tree of SLC22 members implies function. All known mouse and human SLC22 sequences, excluding Slc22a18, were aligned using ICM-Pro v3.8-7c sequence similarity-based alignment. (**A**) Full sequence. (**B**) Extracellular loop (not including Slc22a18, due to its lack of a characteristic large extracellular loop between TMD1 and TMD2). (**C**) Intracellular loop.

**Figure 3 ijms-21-01791-f003:**
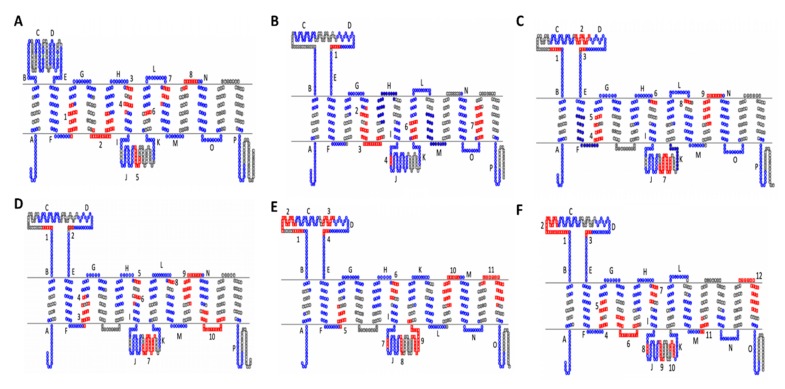
Evolutionarily conserved motifs for each subgroup within the OAT major subgroup mapped onto 2D topology of prototypical members. (**A**) OATS1 mapped onto SLC22A6 (OAT1). (**B**) OATS2 mapped onto hSLC22A7 (OAT2). (**C**) OATS3 mapped onto hSLC22A12 (URAT1). (**D**) OATS3 mapped onto mSlc22a22 (OAT-PG). (**E**) OATS4 mapped onto hSLC22A9 (OAT7). (**F**) OATS4 mapped onto mSlc22a7. In each panel, red sequences are subgroup specific motifs, blue sequences are OAT-major subgroup motifs. Conserved OAT-major subgroup motifs are assigned letters and specific, conserved OAT subgroup motifs are numbered. Data, including motif sequence identities, exact locations, and *p*-values can be found in Appendix A.

**Table 1 ijms-21-01791-t001:** Updated SLC22 family subgroups. The SLC22 family was previously separated into 6 phylogenetic subclades. We propose a reclassification into 8 subgroups based on functional data and supported by the methods described in the text.

Former Groupings		Updated Groupings	
Subclade	Members	Subgroup	Members
OAT	A6, A7, A8, A9, A10, A11, A12, A19, A20, A22, A24, A25, A26, A27, A28, A29, A30	OATS1	A6, A8, A20
		OATS2	A7
		OATS3	A11, A12, a22
		OATS4 ^1^	A9, A10, A24, A25
OAT-like	A13, A14	OAT-like	A13, A14
OAT-related	A17, A18. A23, A31	OAT-related	A17, A18. A23, A31
OCTN-related	A15, A16	OCTN/OCTN related	A4, A5, A15, A16, a21
OCTN	A4, A5		
OCT	A1, A2, A3	OCT	A1, A2, A3

^1^ Six rodent-specific transporters are not included due to their species specificity and lack of functional data.

**Table 2 ijms-21-01791-t002:** Number of SLC22 transporter associations with unique drugs from DrugBank and metabolites. SLC22 transporter substrate specificity (mono-, oligo-, or multi) was predicted from the number of drugs each was associated with. Metabolite data were then used to support the predicted assignment. In the absence of drug data, metabolites were used to determine specificity. #: number, n/a: not applicable.

SLC22 Transporter	Common Name	# of Unique Drugs	# of Metabolites	Sum	Specificity	Metabolic Pathways
A1	OCT1	70	15	85	multi	Monoamines, carnitines, PG ^1^
A2	OCT2	84	24	108	multi	Monoamines, carnitines, PG, creatinine
A3	OCT3	40	12	52	oligo	Monoamines, carnitines, creatinine
A4	OCTN1	33	25	58	oligo	Carnitines, ergothioneine
A5	OCTN2	55	20	75	oligo	Carnitines
A6	OAT1	99	52	151	multi	Uric acid, PG, gut microbiome derived products, TCA ^2^
A7	OAT2	35	16	51	oligo	Cyclic nucleotides, PG, carnitine, creatinine, TCA
A8	OAT3	126	88	214	multi	Uric acid, PG, creatinine, gut microbiome derived products, TCA, bile acids
A9	OAT7	0	9	9	oligo	Conjugated sex steroids, SCFA ^3^
A10	OAT5	3	2	5	mono	Conjugated sex steroids
A11	OAT4	42	9	51	oligo	Uric acid, PG, conjugated sex steroids
A12	URAT1	4	7	11	mono	Uric acid, TCA
A13	OAT10, ORCTL3	n/a	13	13	mono	Uric acid, TCA
A14	ORCTL4	n/a	n/a	n/a	n/a	Understudied
A15	FLIPT1	n/a	7	7	mono	EGT, complex lipids
A16	FLIPT2, CT2	2	16	18	oligo	Carnitines, EGT
A17	BOCT1, NGAL, Lcn2-R	n/a	2	2	mono	Lipocalin
A18	SLC22A1L, TSSC5,	n/a	2	2	n/a	Understudied
A20	OAT6	n/a	13	13	oligo	Odorants, SCFA
a21	Octn3, Slc22a9	n/a	1	1	mono	Carnitine
a22	OAT-PG	n/a	12	12	mono	PG, conjugated sex steroids
A23	BOCT2	n/a	12	12	oligo	Fatty acids
A24	n/a	n/a	10	10	oligo	Conjugated sex steroids, bile acids
A25	UST6	n/a	1	1	mono	Conjugated sex steroids
A31	n/a	n/a	n/a	n/a	n/a	Understudied

^1^ prostaglandins; ^2^ citric acid cycle intermediates; ^3^ short chain fatty acids.

**Table 3 ijms-21-01791-t003:** Genomic localization and tissue expression of the SLC22 family. The following table describes the genomic localization and tissue expression patterns of all SLC22 members excluding the mouse-specific Slc22a19, Slc22a26, Slc22a27, Slc22a28, Slc22a29, and Slc22a30. Slc22a22 and Slc22a21 expression patterns described are from mouse [33,34]. (m) denotes expression patterns observed exclusively in mice. Tissue expression data in humans were collected from various sources and databases [4,24,34]. Expression is assumed from mRNA expression analysis, unless confirmed experimentally. A checkmark represents the presence of the specific transporter in a specific tissue. n/a: not applicable.

			Genomic Loci		Tissue Expression									
Subgroup	SLC22 Transporter	Common Name	Human Chr.	Mouse Chr.	Liver	Kidney	Brain	Gut	Heart	Lung	Testis	Immune Cell	Bone Marrow	Placenta
OATS1	SLC22A6	OAT1	11	19		✓	✓							
	SLC22A8	OAT3	11	19		✓	✓							
	SLC22A20	OAT6	11	19			✓				✓(m)		✓	
OATS2	SLC22A7	OAT2	6	17	✓	✓					✓			
OATS3	SLC22A11	OAT4	11	n/a		✓								✓
	SLC22A12	URAT1	11	19		✓								
	Slc22a22	OAT-PG	-	15		✓(m)								
OATS4	SLC22A9	OAT7	11	-	✓		✓						✓	
	SLC22A10	OAT5	11	-	✓		✓							
	SLC22A24	n/a	11	-	✓	✓	✓	✓			✓			
	SLC22A25	UST6	11	-	✓									
OAT-like	SLC22A13	OAT10, ORCTL3	3	9		✓	✓	✓	✓					
	SLC22A14	ORCTL4	3	9		✓	✓				✓			
OAT-related	SLC22A17	BOCT1, NGAL, Lcn2-R	14	14	✓	✓	✓	✓	✓		✓			
	SLC22A18	SLC22A1L, TSSC5	11	7	✓	✓	✓	✓						
	SLC22A23	BOCT2	6	13	✓	✓	✓	✓		✓				
	SLC22A31	n/a	16	-						✓	✓			
OCTN/OCTN-related	SLC22A4	OCTN1	5	11	✓	✓	✓	✓		✓			✓	
	SLC22A5	OCTN2	5	11	✓	✓	✓	✓		✓		✓		✓
	SLC22A15	FLIPT1	6	10		✓	✓	✓	✓	✓	✓	✓	✓	
	SLC22A16	FLIPT2, CT2	1	3	✓	✓	✓				✓	✓		
	Slc22a21	Octn3, Slc22a9	-	11		✓	✓	✓			✓			✓
OCT	SLC22A1	OCT1	6	17	✓	✓	✓	✓	✓	✓		✓		✓
	SLC22A2	OCT2	6	17	✓	✓	✓	✓		✓				
	SLC22A3	OCT3	6	17	✓	✓	✓	✓	✓	✓		✓		✓

**Table 4 ijms-21-01791-t004:** Combined functional data for OATS4. These data were manually curated and collected from genome-wide association, in vitro, and in vivo studies. Only statistically significant results from each study are included. Column A is the SLC22 transporter, column B is the metabolite, column C is the source of these data (rsid for GWAS, cell line for in vitro, and the physiological measurement for in vivo), column D is the quantitative metric (p value for GWAS, Km, Ki, IC50, or inhibition percentage compared to control for in vitro, and *p* value for in vivo), and column E is the citation.

Gene	Metabolite	Source	Metrics	Citation
SLC22A9	butyrate	in vitro, *Xenopus* oocytes	trans-stimulates transport *p* < 0.001	[37]
SLC22A9	dehydroepiandrosterone sulfate	in vitro, *Xenopus* oocytes	Km: 2.2 uM	[37]
SLC22A9	estrone sulfate	in vitro, *Xenopus* oocytes	Km: 8.7 uM	[37]
SLC22A9	etiocholanolone glucuronide	GWAS, rs113747568	*p* = 5.27 × 10^−28^	[47]
SLC22A9	nicotinate	in vitro, *Xenopus* oocytes	trans-stimulates transport *p* < 0.01	[37]
SLC22A9	progesterone	GWAS, rs112295236	*p* = 8.00 × 10^−12^	[47]
SLC22A9	propionate	in vitro, *Xenopus* oocytes	trans-stimulates transport *p* < 0.01	[37]
SLC22A9	tyramine o-sulfate	GWAS, rs397740636	*p* = 2.06 × 10^−6^	[47]
SLC22A9	valerate	in vitro, *Xenopus* oocytes	trans-stimulates transport *p* < 0.001	[37]
SLC22A10	epiandrosterone sulfate	GWAS, rs1939769	*p* = 2.06 × 10^−7^	[37]
SLC22A10	etiocholanolone glucuronide	GWAS, rs112753913	*p* = 1.88 × 10^−27^	[47]
SLC22A24	androstanediol glucuronide	in vitro, HEK293 Flp-In	IC50: 21 ± 11 uM	[48]
SLC22A24	chenodeoxycholate	in vitro, HEK293 Flp-In	IC50: 2.6 ± 1.0 uM	[48]
SLC22A24	estradiol glucuronide	in vitro, HEK293 Flp-In	3-5 fold over vector control	[48]
SLC22A24	estrone sulfate	in vitro, HEK293 Flp-In	5-10 fold over vector control	[48]
SLC22A24	etiocholanolone glucuronide	in vitro, HEK293 Flp-In	IC50: 29 ± 4.7 uM	[48]
SLC22A24	etiocholanolone glucuronide	GWAS, rs113532193	*p* = 5.90 × 10^−37^	[47]
SLC22A24	pregnanediol-3-glucuronide	in vitro, HEK293 Flp-In	IC50: >200 uM	[48]
SLC22A24	pregnanediol-3-glucuronide	GWAS, rs202187460	*p* = 5.91 × 10^−7^	[47]
SLC22A24	pregnenolone sulfate	in vitro, HEK293 Flp-In	IC50: 1.4 ± 0.1 uM	[48]
SLC22A24	progesterone	in vitro, HEK293 Flp-In	IC50: 7.4 ± 3.0 uM	[48]
SLC22A24	taurocholic acid	in vitro, HEK293 Flp-In	10–20 fold over vector control	[48]
SLC22A24	ursodeoxycholate	in vitro, HEK293 Flp-In	IC50: 7.6 ± 1.2 uM	[48]
SLC22A25	etiocholanolone glucuronide	GWAS, rs113950742	*p* = 4.12 × 10^−27^	[47]

**Table 5 ijms-21-01791-t005:** ICM finds significant similarities with SLC22 members. The following table shows significant amino acid similarities found between full-length and the ECD sequences of SLC22 members and other known proteins from human (*Homo sapiens*, h), cow (*Bos taurus*, b), chicken (*Gallus gallus*, g), mouse (*Mus musculus*, m), and rat (*rattus norvegicus*, r). No significant similarities were found for SLC22 ICDs. pP value is the log of the *p*-value and is described in the methods.

Subclade	SLC22 Family Member	Common Name	Non-SLC22 Protein	Identity Shared (%)	pP Value
OCT	hSLC22A1 ECD	OCT1	hSCO-spondin	28.97	5.47
			bSCO-spondin	30.84	5.35
			mSCO-spondin	24.3	6.24
			rSCO-spondin	24.3	6.16
	hSLC22A2 ECD	OCT2	bSCO-spondin	30.84	5.29
			mSCO-spondin	25.23	5.92
			rSCO-spondin	24.3	5.61
	hSLC22A3 ECD	OCT3	hSCO-spondin	27.43	5.33
			bSCO-spondin	22.12	5.49
			mSCO-spondin	27.43	5.89
			rSCO-spondin	25.66	5.81
OAT-related	hSLC22A31 ECD	n/a	hRBM42	30.95	5.95
			bRBM42	32.14	6.05
			mRBM42	30.95	5.95
			rRBM42	30.95	5.95
			hBAHD1	36.9	5.39
OCTN	mSlc22a16 ECD	FLIPT2, CT2	gCRBB3	26	5.4
	hSLC22A16		hTAS2R41	20	5.3
	hSLC22A5	OCTN2	GPR160	21	6.1

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
