# Peer review of "Systems Biology Analysis Reveals Eight SLC22 Transporter Subgroups, Including OATs, OCTs, and OCTNs"

_ijms, 2020, doi:10.3390/ijms21051791_

Round 1

Reviewer 1 Report

The authors of the article carry out a systematic study on the SLC22 transporter family which presents itself as a class of transport proteins with a central role in various physiological aspects. Although it is known about these proteins as a drug transporters, however their role is much wider it's interesting. They present a wide and detailed treatment including tables, multiple alignments, genomic localization, tissue expression, 2D topological models that make the work very complete and exhaustive.The text is presented in a clear, complete form that scientifically offers a contribution to understanding the physiological role of these proteins. The authors highlight an important concept related to the specificity and multi-specificity of this class of proteins that make them a very important study object to better understand the metabolic network in which they are involved making this study approach applicable to many other proteins. Although some minor corrections on punctuation and on some tables (eg tab. 3) are required, however, the article can be accepted for publication.  

Author Response

Dear Editors, 

The following changes have been made to the manuscript: Engelhart et. al. Systems Biology Analysis Reveals Eight SLC22 Transporter Subgroups, Including OATs, OCTs, and OCTNs;

The legend for Table 4 has been replaced with “Combined functional data for OATS4. These data were manually curated and collected from genome-wide association, in vitro, and in vivo studies. Only statistically significant results from each study are included. Column A is the SLC22 transporter, column B is the metabolite, column C is the source of the data (rsid for GWAS, cell line for in vitro, and the physiological measurement for in vivo), column D is the quantitative metric (p value for GWAS, Km, Ki, IC50, or inhibition percentage compared to control for in vitro, and p value for in vivo), and column E is the citation.”

Reviewer 1:

The authors of the article carry out a systematic study on the SLC22 transporter family which presents itself as a class of transport proteins with a central role in various physiological aspects. Although it is known about these proteins as a drug transporters, however, their role is much wider it's interesting. They present a wide and detailed treatment including tables,multiple alignments, genomic localization, tissue expression, 2D topological models that make the work very complete and exhaustive. The text is presented in a clear, complete form that scientifically offers a contribution to understanding the physiological role of these proteins. The authors highlight an important concept related to the specificity and multi-specificity of this class of proteins that make them a very important study object to better understand the metabolic network in which they are involved making this study approach applicable to many other proteins. Although some minor corrections on punctuation and on some tables (eg tab. 3) are required, however, the article can be accepted for publication. 

In Table 3, “SL22A6” was changed to “SLC22A6”.

Reviewer 2:        

This manuscript reports on a reassessment of the classification of the SLC22 group of transporters. This analysis is very thorough and will provide a useful addition to researchers interested in these genes. The manuscript is quite lengthy and a general comment is that it can be difficult to keep track of the different groups since the numerical order (SLC22A1 -SLC22A31) does not align with the new groupings (or for that matter, the old), eg A6, A8 and A20 fall within the new OATS1 subgroup. The authors also seem to prefer introducing the transporters in some sections with their SLC22 nomenclature but revert to using the common name in the remainder of the paragraph/section adding to the difficulty of keeping track of the group being discussed, eg section 2.3. Perhaps one day a more logical renaming of this fascinating group of transporters can take place.

  1. The word “data”. Generally, scientific writing should use the plural form, ie data are, these data, and I note an inconsistent use throughout the manuscript.

The following instances of data were changed.

Line 101 “Emerging data continues” was changed to “emerging data continue”

Line 208 “Tissue expression data in humans was collected” was changed to “Tissue expression data in humans were collected”

Line 263 “Transport data for Slc22a9/a24 shows shared” was changed to “Transport data for Slc22a9/a24 show shared”

Line 286 “Very little functional data is available” was changed to “Very little functional data are available

Line 290 “this data” was changed to “these data” 

Line 341 “GWAS data suggests” was changed to “GWAS data suggest”

Line 343 “Although data is very limited” was changed to “Although data are very limited”

Line 494 “Although the functional data is inherently biased” was changed to “Although the functional data are inherently biased”

Line 496 “there is enough data” was changed to “there are enough data”

Line 521 “Existing functional data suggests” was changed “Existing functional data suggest”

Line 592 “Functional data has come” was changed to “Functional data have come”

Line 663 “Most in vitro data comes from” was changed to “Most in vitro data come from”

Line 665 “Metabolite data was” was changed to “Metabolite data were”

  1. Introduction: Line 77 Perhaps a brief mention of what the new data are; functional? genomic? phylogenetic? Expression?

Line 78 “data” changed to “functional data”

  1. Results: Sections are initially labelled 2.x, but then revert to 1.x at 1.4 onwards.

Line 213 “1.4” changed to “2.4”

Line 232 “1.5” changed to “2.5”

Line 247 “1.6” changed to “2.6”

Line 285 “1.7” changed to “2.7”

Line 299 “1.8” changed to “2.8”

Line 318 “1.9” changed to “2.9”

Line 332 “1.10” changed to “2.10”

Line 353 “1.11” changed to “2.11”

Line 388 “1.12” changed to “2.12”

Line 407 “1.13” changed to “2.13”

Line 454 “1.13” changed to “2.14”

  1. Six transporters are rodent specific and are omitted from “Updated Groupings” in Table 1. However, this omission is not explained until much later; in section 1.6. It would be more useful to comment on the rodent specific transporters early on as a reader less familiar with this SLC22 family will wonder where and why those six proteins were omitted. Perhaps a footnote to Table 1 to explain their fate can be added?

Footnote added to table 1 stating why rodent transporters are not included: “Six rodent-specific transporters are not included due to their species specificity and lack of functional data”

  1. Line 269 Table 4 legend is repeat of Table 3’s.

The Table 4 legend has been updated to reflect its content.

  1. Figure 3 The assigned letters for subgroup motifs are too small for a printed version. The green diamonds marking SNPs appear invisible to me even with high magnification on screen.

Size of Figure 3 was increased and mention of green diamonds were removed. 

  1. Table 5 shows SCO-spondins sharing motifs with OCTs but these are not discussed in the text. Some mention of the function of the common motif should be included.

Line 460-465: Information about SCO-spondin included in the text.

Thank you for your time, 

Darcy Engelhart 

Reviewer 2 Report

Englehart et. al. Systems Biology Analysis Reveals Eight SLC22 Transporter Subgroups, Including OATs, OCTs, and OCTNs.

This manuscript reports on a reassessment of the classification of the SLC22 group of transporters. This analysis is very thorough and will provide a useful addition to researchers interested in these genes. The manuscript is quite lengthy and a general comment is that it can be difficult to keep track of the different groups since the numerical order (SLC22A1 -SLC22A31) does not align with the new groupings (or for that matter, the old), eg A6, A8 and A20 fall within the new OATS1 subgroup. The authors also seem to prefer introducing the transporters in some sections with their SLC22 nomenclature but revert to using the common name in the remainder of the paragraph/section adding to the difficulty of keeping track of the group being discussed, eg section 2.3. Perhaps one day a more logical renaming of this fascinating group of transporters can take place.

The word “data”. Generally, scientific writing should use the plural form, ie data are, these data, and I note an inconsistent use throughout the manuscript.

Introduction

Line 77 Perhaps a brief mention of what the new data are; functional? genomic? phylogenetic? expression?

Results.

Sections are initially labelled 2.x, but then revert to 1.x at 1.4 onwards.

Six transporters are rodent specific and are omitted from “Updated Groupings” in Table 1. However, this omission is not explained until much later; in section 1.6. It would be more useful to comment on the rodent specific transporters early on as a reader less familiar with this SLC22 family will wonder where and why those six proteins were omitted. Perhaps a footnote to Table 1 to explain their fate can be added?

Line 269 Table 4 legend is repeat of Table 3’s.

Figure 3 The assigned letters for subgroup motifs are too small for a printed version. The green diamonds marking SNPs appear invisible to me even with high magnification on screen.

Table 5 shows SCO-spondins sharing motifs with OCTs but these are not discussed in the text. Some mention of the function of the common motif should be included.

Author Response

(The authors gave the same response as above.)
